# Exploring How Cyanobacterial Traits Affect Nutrient Loading Thresholds in Shallow Lakes: A Modelling Approach

**Manqi Chang [1,2,*]** , **Sven Teurlincx [1]** , **Jan H. Janse [1,3]** , **Hans W. Paerl [4]** , **Wolf M. Mooij [1,2]** and **Annette B. G. Janssen [5]**

1. Department of Aquatic Ecology, Netherlands Institute of Ecology (NIOO-KNAW), P.O. Box 50, 6700 AB Wageningen, The Netherlands; s.teurlincx@nioo.knaw.nl (S.T.); jan.janse@pbl.nl (J.H.J.); w.mooij@nioo.knaw.nl (W.M.M.)
2. Department of Aquatic Ecology and Water Quality Management, Wageningen University & Research, P.O. Box 47, 6700 AA Wageningen, The Netherlands
3. PBL, Netherlands Environmental Assessment Agency, P.O. Box 30314, 2500 GH The Hague, The Netherlands
4. Institute of Marine Sciences, The University of North Carolina at Chapel Hill, Morehead City, NC 28557, USA; hans_paerl@unc.edu
5. Water Systems and Global Change Group, Wageningen University & Research, P.O. Box 47, 6700 AA Wageningen, The Netherlands; annette.janssen@wur.nl
* Correspondence: m.chang@nioo.knaw.nl

**Abstract:** Globally, many shallow lakes have shifted from a clear macrophyte-dominated state to a turbid phytoplankton-dominated state due to eutrophication. Such shifts are often accompanied by toxic cyanobacterial blooms, with specialized traits including buoyancy regulation and nitrogen fixation. Previous work has focused on how these traits contribute to cyanobacterial competitiveness. Yet, little is known on how these traits affect the value of nutrient loading thresholds of shallow lakes. These thresholds are defined as the nutrient loading at which lakes shift water quality state. Here, we used a modelling approach to estimate the effects of traits on nutrient loading thresholds. We incorporated cyanobacterial traits in the process-based ecosystem model PCLake+, known for its ability to determine nutrient loading thresholds. Four scenarios were simulated, including cyanobacteria without traits, with buoyancy regulation, with nitrogen fixation, and with both traits. Nutrient loading thresholds were obtained under N-limited, P-limited, and colimited conditions. Results show that cyanobacterial traits can impede lake restoration actions aimed at removing cyanobacterial blooms via nutrient loading reduction. However, these traits hardly affect the nutrient loading thresholds for clear lakes experiencing eutrophication. Our results provide references for nutrient loading thresholds and draw attention to cyanobacterial traits during the remediation of eutrophic water bodies.

**Keywords:** harmful algal blooms; regime shift; alternative stable state; resilience; hysteresis; light limitation; nutrient limitation; critical nutrient loading

## 1. Introduction

Globally, many shallow lakes have shifted from a clear macrophyte-dominated state to a turbid phytoplankton-dominated state due to eutrophication [1–3]. Such a shift in water quality state is often accompanied by toxic cyanobacterial blooms and their subsequent nuisance properties [4,5]. Driven by wind, these blooms can float to shoreline areas, threatening drinking water safety and deteriorating

recreational values of lakes [6–8]. Therefore, lake managers aim to restore their lakes to a clear state that can usually provide more societally relevant ecosystem functions compared to a turbid state [9–11]. However, restoration of these lakes is often challenging because of the lack of response of lakes to measures such as reducing nutrients [12].

This lack of response of shallow lakes originates from the positive feedback between macrophytes and phytoplankton, resulting in resistance to external pressures such as changing nutrient loading [13–15]. Due to this resistance, a clear or turbid state of a lake can remain itself until a threshold of nutrient loading, hereafter referred to as nutrient loading threshold, is crossed. This nutrient loading threshold is also known as the critical nutrient loading [16–20]. In shallow lake ecosystems, there are usually two values of nutrient loading thresholds. One value is identified during the process of eutrophication when a lake shifts from clear to turbid. This value can assist managers and decision-makers with setting limits on nutrient loading to prevent a lake shifting to a turbid state with a typically inevitable loss of ecosystem services [11]. The other value is identified during the process of oligotrophication when a lake shifts from a turbid to a clear state. This value can be used to set the target of nutrient loading reduction in lake restoration actions aiming at improving the water quality of a eutrophic lake [17]. Cyanobacteria as the dominant species in eutrophic shallow lakes have the potential to challenge water quality improvement [21]. They have benefited from a long evolutionary history during which traits have developed and provided them with competitive advantages such as buoyancy regulation and nitrogen fixation [22]. These traits promote access to light and nutrients accessibility for cyanobacteria and can have impacts on lake nutrient loading thresholds.

Buoyancy regulation is a trait that allows cyanobacteria to rapidly adjust their vertical position in the water column [23,24]. The advantage of buoyancy regulation for cyanobacteria is well recognized, as it supports cyanobacterial biomass by providing access to light at the surface and outcompetes the other phytoplankton species through shading [5,25,26]. Buoyancy regulation is achieved in two ways that involve cell- and colony-level processes. At the cellular level, buoyancy regulation is provided by gas vesicles in cells of some cyanobacterial species, e.g., *Microcystis, Dolichospermum* (formerly *Anabaena*), and *Oscillatoria* [27]. These gas vesicles allow cyanobacteria cells to have a lower density than water so that they can float [27]. At a colonial level, buoyancy regulation is enhanced by a colony formation that enlarges the colony radius, thereby increasing the buoyancy [28]. These colonies with high buoyancy can persist at the surface for lengthy periods and enable cyanobacteria to become dominant, especially in warm, eutrophic and stagnant water bodies [22,29].

Nitrogen fixation allows cyanobacteria to utilize atmospheric nitrogen as a source of nitrogen. Therefore, nitrogen-fixing cyanobacteria have an advantage over non-nitrogen fixers when nitrogen is deficient [30], which occurs when nitrogen loading is relatively low or when the denitrification rate is high in the water column [25,31,32]. Nitrogen fixation is an energy-consuming process that requires cyanobacteria to utilize sufficient light [26,33]. The aforementioned buoyancy regulation trait provides cyanobacteria ample access to light, which may then facilitate energy-intensive nitrogen fixation. Examples of cyanobacterial genera known for nitrogen fixation are *Dolichospermum* and *Aphanizomenon* [34,35].

Previous work has mainly focused on how these traits contribute to cyanobacterial competitive strength [5,24,28,36], yet there is little known about how these traits affect the value of nutrient loading thresholds of shallow lakes. The values of nutrient loading threshold are typically obtained using a modelling approach [17], since studies on critical transitions based on field data are rare due to the requirements of long time series [37–40] and mesocosm experiments cannot represent the field situation [41–43]. Moreover, models can analyse the nutrient loading threshold for almost unlimited scenarios without disturbing the lake. Cottingham et al. [44] used a simple mathematical model to show that cyanobacterial traits can reduce lake resistance to external nutrient loading and accelerate eutrophication. Yet, their approach lacks real ecosystem processes such as food web interactions and competition between phytoplankton and macrophytes, thus making a useful evaluation of the effect of cyanobacterial traits on nutrient loading thresholds difficult. For this purpose, a process-based model

that incorporates biogeochemical processes is needed as a good tool to study the effect of cyanobacterial traits on the value of nutrient loading threshold [17,19].

In this study, we incorporated established knowledge from empirical research on buoyancy regulation and nitrogen fixation into a process-based ecosystem model. Then, to account for different combinations of cyanobacterial traits, we analysed four conceptual scenarios where cyanobacteria have no traits, one of the two traits, or both traits, respectively. Because different N:P loading ratios lead to different nutrient limitations [45], each scenario was simulated with a low (nitrogen-limited), a high (phosphorus-limited) and an intermediate (colimited) N:P weight ratios. Through this modelling effort, we evaluated how the cyanobacterial traits of buoyancy regulation and nitrogen fixation affect nutrient loading thresholds during lake eutrophication and oligotrophication.

## 2. Methods

### 2.1. Model Selection

We used a modelling approach to explore the effect of cyanobacterial traits on nutrient loading thresholds at the ecosystem level. We chose to use PCLake+ [46], which is the updated version of PCLake [47], as the model is known for its ability to calculate nutrient loading thresholds. PCLake+ was deemed suitable for our study for four reasons. First, PCLake+ can represent real lake ecosystems because it incorporates complex biogeochemical processes, including nutrient cycling, phytoplankton species and macrophyte competition, and zooplankton and fish food web interactions. Second, processes and interactions of cyanobacteria developed in this model have been validated [17,47,48]. Third, PCLake+ can be used in bifurcation analyses to identify nutrient loading thresholds during lake eutrophication and oligotrophication [17,19,47,49]. Finally, DATM (Database Approach To Modelling) provides an interface where the framework of equations used in PCLake+ is explicitly presented. This explicit framework enabled us to easily incorporate new processes by changing or adding equations to the model [50,51].

### 2.2. Model Adaptation

Using established knowledge from empirical research, we expanded PCLake+ by adding and modifying equations for the cyanobacterial traits buoyancy regulation and nitrogen fixation.

#### 2.2.1. Buoyancy Regulation

To incorporate buoyancy regulation in PCLake+, we coded an additional surface layer on the top of the existing epilimnion layer. Moreover, we developed a new state variable for cyanobacteria that are present in the newly developed surface layer (hereafter referred to as "surface cyanobacteria"). The equations for growth and mortality processes of the surface cyanobacteria were coded similarly to the original equations of cyanobacteria in the epilimnion. The equation for light intensity in the water column was adjusted to account for extinction caused by surface cyanobacteria. As a result, surface cyanobacterial blooms can shade macrophytes and phytoplankton species such as chlorophytes and diatoms while competing for nutrients in the water column. In addition, we included an equation that accounts for reduced oxygen aeration rate when dense surface cyanobacteria hinder the interactions between water and atmosphere [52]. This process was included in a similar method as the equation applied for the floating plant *Lemna* in PCDitch [18,53].

Empirical research shows that buoyancy regulation by cyanobacteria depends on wind, temperature, and biomass conditions [54,55]. Strong winds disable the formation of floating layers. Studies have shown that wind speed above a certain threshold causes turbulence that can disperse cyanobacterial colonies, thereby impeding buoyancy [56,57]. Temperature provides energy for cyanobacteria to produce the protein-forming gas vesicles [58]. These gas vesicles can lower the cyanobacterial cell density below water density, enabling cyanobacteria to float [58,59]. Conversely, cyanobacterial cells start settling when the temperature is too low to support gas vesicle production,

such as during autumnal sedimentation [60]. High levels of cyanobacterial biomass can indirectly facilitate buoyancy regulation because of colony formation [25,61]. Cyanobacterial colonies have higher buoyancy compared to individual cells [62], enabling cyanobacterial colonies to remain at the surface for a longer period during summer compared to noncolony-forming species [56,63].

For these three processes, we defined the actual floating speed $F_f$ (m d$^{-1}$) of the planktonic cyanobacteria in PCLake+ by using a maximum floating speed $F_{max}$ (m d$^{-1}$) times three limitation factors for wind speed ($\sigma_W$), temperature ($\sigma_T$), and biomass ($\sigma_B$), as shown in Equation (1).

$$F_f = F_{max}\, \sigma_W\, \sigma_T\, \sigma_B \tag{1}$$

Each limiting factor was defined as a Hill equation, following a similar concept of a step function as used in the book by Alon [64] (Equations (2)–(4)),

$$\sigma_W = 1 - \frac{W^b}{H_W^b + W^b} \tag{2}$$

$$\sigma_T = \frac{T^b}{H_T^b + T^b} \tag{3}$$

$$\sigma_B = \frac{B^b}{H_B^b + B^b} \tag{4}$$

in which $W$, $T$, and $B$ are wind speed, temperature, and biomass, respectively; $\sigma_W$, $\sigma_T$, and $\sigma_B$ are limitation factors for floating caused by wind, temperature, and biomass, respectively; transition constants $H_W$, $H_T$, and $H_B$ were defined to mark the transition between limited and unlimited conditions for wind speed, temperature, and biomass, respectively; and $b$ is the coefficient that defines the smoothness of the Hill equation. Values of Equations (2)–(4) vary between zero and one where a value of zero denotes serious limitation and one denotes no limitation. For example, the wind limitation factor (Equation (2)) will approach zero if the wind speed $W$ exceeds the transition constant $H_W$. Similarly, buoyancy regulation becomes limited if the temperature and biomass are below their transition constants (note the sign of b).

Following empirical research (see for references Table 1), we set the transition constants for wind at 3 m s$^{-1}$, for temperature as 15 °C, and for biomass at 10 mg chl-*a* m$^{-3}$ (Table 1). We selected 10 m day$^{-1}$ as the maximum floating speed (Table 1). Both biomass and temperature are dynamically calculated in the model. We applied a constant wind speed of 2 m s$^{-1}$, which is lower than the transition constant so that the wind will not hinder buoyancy regulation in this study. Net floating speed $F_{net}$ (m day$^{-1}$) was defined by an actual floating speed $F_f$ (m day$^{-1}$) minus the settling speed $S$ (m day$^{-1}$) that depends on resuspension and temperature (Equation (5)).

$$F_{net} = F_f - S \tag{5}$$

**Table 1.** Applied transition constants and maximum floating speed for the calculation of actual floating speed of cyanobacteria in PCLake+.

| Parameter | Description | Reference Value | Applied Value | Unit | Model Notation |
|---|---|---|---|---|---|
| $H_W$ | Transition constant of wind speed | 3–4 [55,65–67], 2 [68,69], 2–3 [56], 4 [28] | 3 | m s$^{-1}$ | cVWindThrBlue |
| $H_T$ | Transition constant of temperature | 12–18 [60,70] | 15 | °C | cTmThrBlue |
| $H_B$ | Transition constant of cyanobacterial biomass (expressed as chl-*a* level) | 10–100 [55], 100 [71], 20 [16] | 10 | mg chl-*a* m$^{-3}$ | cChlaThrBlue |
| $W_{max}$ | Maximum floating speed | 1–10 [71], 4.5 [72], 6 [73], 0.2–250 [74], 10 [75] | 10 | m day$^{-1}$ | cVFloMaxBlueW |

### 2.2.2. Nitrogen Fixation

Besides available $NH_4^+$ and $NO_3^-$ in the water column, nitrogen fixation generates additional nitrogen loading from the atmosphere that is neglected in the original PCLake+ model. Here, we developed equations to account for these extra sources of nitrogen in terms of dissolved $N_2$ to planktonic and benthic cyanobacteria. Therefore, cyanobacteria with nitrogen fixation in the expanded PCLake+ can use the nitrogen pools of $NH_4^+$, $NO_3^-$, and dissolved $N_2$.

Many empirical studies have shown that nitrogen fixation requires substantial energy and hence is dependent on light intensity [35,76,77]. For this positive relationship between light intensity and nitrogen fixation rates, we used the model from Ferber et al. [32], which expanded upon the model by Levine et al. [78], as shown in Equations (6) and (7):

$$N_i = N_S - N_S e^{-a} + D \tag{6}$$

$$a = \alpha N_S^{-1} I_{ave} \tag{7}$$

in which $N_i$ ($\mu$gN mgChl$^{-1}$ day$^{-1}$) is the fixation of nitrogen at certain light intensity $I_{ave}$ (J m$^{-2}$ s$^{-1}$), $N_S$ ($\mu$gN mgChl$^{-1}$ day$^{-1}$) is fixation at light saturation, $D$ ($\mu$gN mgChl$^{-1}$ day$^{-1}$) is fixation in the dark, and $\alpha$ is the slope of the rising limb of the relationship between fixation and light intensity. The parameter units used in Ferber et al. [32] were converted to the parameter units used in PCLake+. To simplify the calculation of $a$ (Equation (7)), we combined $\alpha N_S^{-1}$ ((J m$^{-2}$ s$^{-1}$)$^{-1}$) as one constant parameter. Original and applied values of parameters are provided in Table 2. In addition, the light intensity $I_{ave}$ was calculated by averaging the integral of light intensity through depth according to the Beer–Lambert law:

$$I_{ave} = \frac{I_{in} - I_{in} e^{-kZ}}{kZ} \tag{8}$$

in which $I_{in}$ (J m$^{-2}$ s$^{-1}$) is the light intensity at the surface, $k$ (m$^{-1}$) is the total light extinction in the water column, and $Z$ (m) is the lake depth.

**Table 2.** Parameter values adopted in PCLake+ used for the nitrogen fixation process. The parameter units used in Ferber et al. [32] were converted to be the parameter units that can be coupled with current processes in PCLake+.

| Parameter | Description | Reference Value * | Applied Value | Model Notation |
|---|---|---|---|---|
| $N_S$ | Nitrogen fixation rate at light saturation | 366 nmol $N_2$ ($10^6$ heterocysts)$^{-1}$ h$^{-1}$ | 60 $\mu$gN mgChl$^{-1}$ day$^{-1}$ | cNfixMaxBlue |
| $D$ | Nitrogen fixation rate in the dark | 58 nmol $N_2$ ($10^6$ heterocysts)$^{-1}$ h$^{-1}$ | 9.5 $\mu$gN mgChl$^{-1}$ day$^{-1}$ | cNfixDarkBlue |
| $\alpha N_S^{-1}$ | Auxiliary parameter for nitrogen fixation–light curve | 2.16 (Ei m$^{-2}$)$^{-1}$ | 0.036 (J m$^{-2}$ s$^{-1}$)$^{-1}$ | cAlphNfix |
| $I_{ave}$ | The average of integral light through depth | dynamically calculated (Ei m$^{-2}$) | dynamically calculated (J m$^{-2}$ s$^{-1}$) | aLPARAveSurf, aLPARAveEpi, aLPARAveHyp |

\* For the conversion method from reference value to applied value see Supplementary material 1.

### 2.3. Bifurcation Analyses

To study the effect of buoyancy regulation and nitrogen fixation on the nutrient loading thresholds, we used the bifurcation analyses [17,79] with the expanded version of PCLake+. A bifurcation analysis is a mathematical tool to study large sudden changes in the output (here chl-*a*) as a result of small changes in the input (here nutrient load). A large sudden change indicates a nutrient loading threshold. Using bifurcation analyses to identify nutrient loading thresholds require several steps. Firstly, a range of nutrient loadings along a wide gradient needs to be defined. Secondly, the model needs to be run to equilibrium for each of these nutrient loadings in the selected range to calculate the chl-*a* at

equilibrium. Thirdly, a curve can be drawn showing the response of chl-*a* levels to nutrient loading using the results of the equilibrium chl-*a* level as the y-axis and the corresponding nutrient loading as the x-axis. With this type of curve, a bifurcation plot can be obtained. Finally, the nutrient loading thresholds can be thus defined as the point of a sudden change in chl-*a* level in the bifurcation plot.

A bifurcation plot for shallow lakes commonly shows two lines (for example, see Janse et al. [19,20]); one line shows the results of simulations starting with a highly oligotrophic macrophyte-dominated state and commonly has a high nutrient loading threshold. The other line shows the results of simulations starting from a highly eutrophic phytoplankton-dominated state and commonly has a low nutrient loading threshold. The initial states needed to calculate these two lines can be set by using the values of state variables that represent an excessively clear and an excessively turbid state, respectively. These values are calibrated by Janse [47]. The line with a high nutrient loading threshold can show a sudden increase of the chl-*a* level during eutrophication and illustrate the process of a lake shifting from a clear to a turbid state. The line with a low nutrient loading threshold can show a sudden decrease of chl-*a* level during oligotrophication and illustrate the process of a lake shifting from a turbid to a clear state.

In our study, each bifurcation analysis consists of 400 simulations for which 200 are used to calculate the oligotrophication line and 200 to calculate the eutrophication line. Both lines are calculated using a range of N-loads with 200 equal intervals between 0.0005 and 0.046 gN m$^{-2}$ day$^{-1}$. We verified that the sudden changes in chl-*a* level are visible within this range of nutrient loadings. Each simulation was run for 50 years to reach the equilibrium. We then calculated the yearly average of the fiftieth year. From the 200 values for yearly averaged chl-*a*, we constructed one line of the bifurcation plot.

We performed the bifurcation analyses for scenarios with different combinations of the traits buoyancy regulation and nitrogen fixation. The four scenarios that we used are (1) a control scenario with neither buoyancy regulation nor nitrogen fixation (CT), (2) a scenario with only buoyancy regulation (BR), (3) a scenario with only nitrogen fixation (NF), and (4) a scenario where the cyanobacteria could both regulate buoyancy and fix nitrogen (BR + NF). We expect that the effects of cyanobacterial traits on the nutrient loading thresholds depend at least in part on the N:P ratio of the incoming load, which in return can lead to different nutrient limitations. We chose to set the N:P ratio at 5 for nitrogen-limitation, 20 for phosphorus-limitation, and 10 for colimitation. We applied the different N:P ratios by dividing the N-loads by 5, 10, or 20, respectively, to set the P-loads. In total, we performed 12 bifurcation analyses to assess all combinations of the four scenarios with the three N:P ratios.

We used the nutrient loading thresholds of the control scenario without traits as a benchmark and calculated its differences with the other three scenarios with traits under different N:P ratios. Besides the typical bifurcation plots that provide the nutrient loading threshold, we also reported phytoplankton composition to interpret the role of cyanobacterial traits in their competition with other algal species for different nutrient loadings. Because there are in total 12 scenarios and each of them requires 400 simulations to obtain a bifurcation plot, an R script was used to automatically activate the expanded PCLake+ under the DATM framework and perform these simulations (the script is shown in Supplementary material 2).

All model analyses performed in this study used a lake depth of 2 m and assumed complete vertical mixing. We note that PCLake+ is provided with an epilimnion and hypolimnion layer that can stratify [46]. We did not use these functionalities for this study because a sudden shift in the water quality state of deep lakes is less common [14]. Nevertheless, we implemented the traits for both layers to cater to future studies of deep lakes. All equations, as well as state and parameter values, are presented in Supplementary material 3. The full model can be downloaded from https://github.com/pcmodel/PCModel/tree/master/Licence_agreement/I_accept/PCModel1350/PCModel/3.00/Models/PCLake+/6.13.16/DATM/Chang_et_al_2020_Water/.

## 3. Results

Bifurcation analyses with the expanded version of PCLake+ showed that cyanobacterial traits affected nutrient loading thresholds (Figure 1). Compared with the control scenario, the results for scenarios with traits show reduced nutrient loading thresholds for the shift from turbid to clear (Figure 2). The reductions in the nutrient loading threshold vary between 12% and 46% (compared to a control where cyanobacteria lack these traits) depending on the limiting nutrient and the different cyanobacterial traits. The NF scenario showed the most reduction in the nutrient loading threshold when the N:P ratio was 5 (Figure 2). The BR+NF scenario showed the most reduction when the N:P ratio was 10. When the N:P ratio was 20, the NF scenario showed no effect on the nutrient loading threshold and the BR and BR + NF gave a comparable reduction in the nutrient loading thresholds during oligotrophication. Additionally, the chl-*a* levels of all scenarios with cyanobacterial traits revealed a different biomass in the turbid state compared to the CT scenario, except for the NF scenario under an N:P ratio of 20, where the two overlapped. Detailed results for each scenario are described in the sections below. The nutrient loading thresholds for the shift from clear to turbid were similar between all the scenarios (Figure 2). This suggests that once the cyanobacteria are dominant, the traits enable cyanobacteria to withstand efforts to eliminate them through nutrient reduction, whereas their emergence during eutrophication is hardly affected.

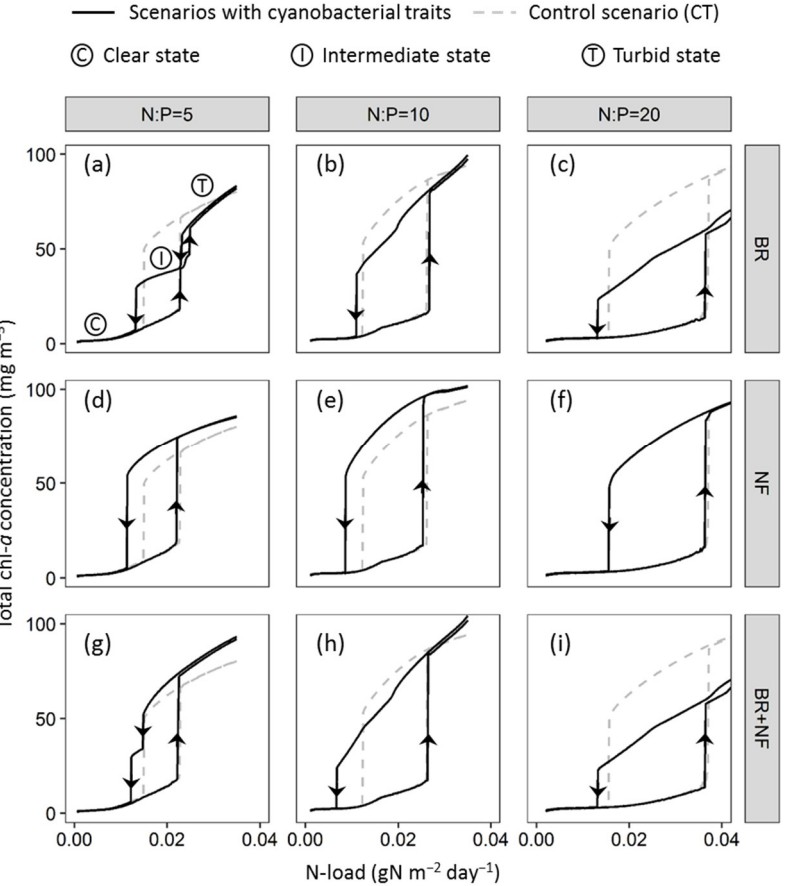

**Figure 1.** Bifurcation plots of total chl-*a* concentration for different scenarios of cyanobacterial traits (black lines) compared to the control (grey dashed lines: default PCLake+). The scenarios shown in the figure are (**a**–**c**) buoyancy regulation (BR), (**d**–**f**) nitrogen fixation (NF), and (**g**–**i**) combination of buoyancy regulation and nitrogen fixation (BR + NF). Simulations started from a turbid initial state are marked by arrows downwards, simulations started from a clear initial state are marked by arrows upwards.

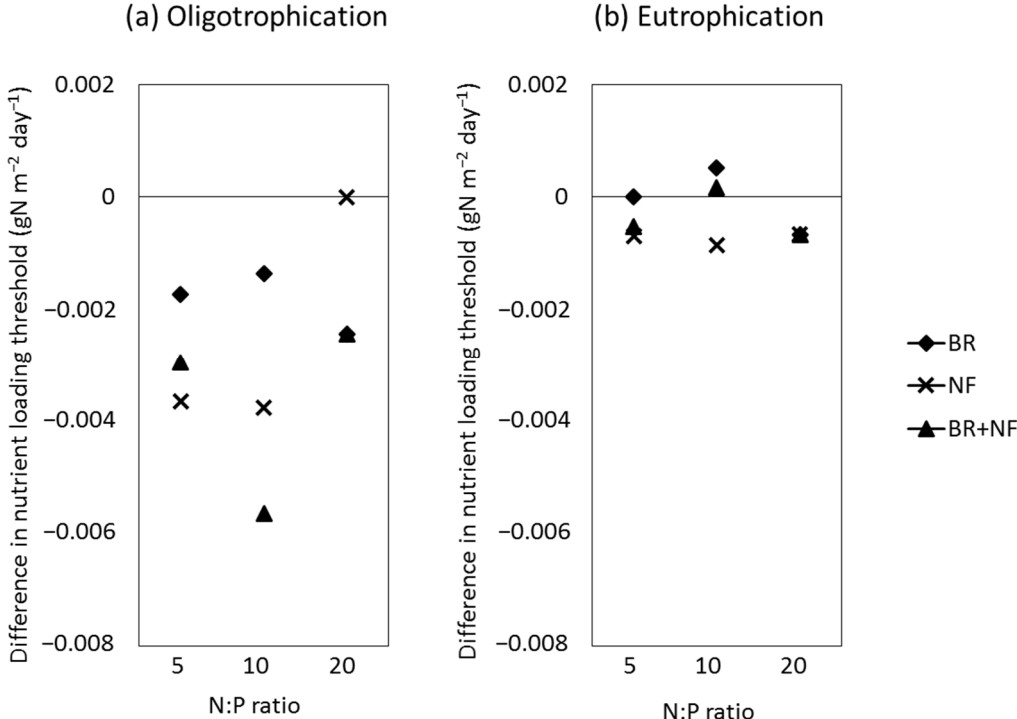

**Figure 2.** Differences in nutrient loading threshold between the control (CT) scenario and scenarios with traits, including the buoyancy regulation (BR) scenario with diamonds, the nitrogen fixation (NF) scenario with crosses, and the combination of buoyancy regulation and nitrogen fixation (BR + NF) scenario with triangles. (**a**) The difference of nutrient loading threshold obtained during oligotrophication and (**b**) the difference of nutrient loading threshold obtained during eutrophication.

## 3.1. Effect of Buoyancy Regulation (BR Scenario)

Inclusion of buoyancy regulation in the model led to a reduction of the chl-*a* levels from 9% to 40% just after the lake shifted from a clear to turbid state (Figure 1a–c). Yet, with increasing nutrient loadings, the chl-*a* levels show a stronger increase in the BR scenario compared with the CT scenario. Consequently, the chl-*a* level of the BR scenario surpassed the chl-*a* level of the CT scenario in the high nutrient loading scenario (see N:P is 5 and 10 in the graph, N:P is 20 and off-graph). For the BR scenario with different N:P ratios, the shape of the bifurcation plots showed three states. The first two are the clear macrophyte-dominated state where chl-*a* level is low and a turbid phytoplankton-dominated state where chl-*a* level is high. These two states are typically seen in shallow lakes [13]. The third is an intermediate state where phytoplankton outcompete macrophytes (Figure S1) and chl-*a* level is intermediately high ("I" in Figure 1a). While the eutrophication line follows the typical pattern of a bifurcation plot of shallow lakes, the oligotrophication line with an N:P ratio of 5 started with a turbid state ("T" in Figure 1a) and shifted to an intermediate state ("I" in Figure 1a) before returning to a clear state ("C" in Figure 1a). Figure 3 demonstrates cyanobacterial dominance in the turbid state while diatoms are dominant in both the intermediate and clear states. Further, surface cyanobacteria were present when the scenario accounted for buoyancy regulation and they had higher biomass than the cyanobacteria in the water column, except during the intermediate state.

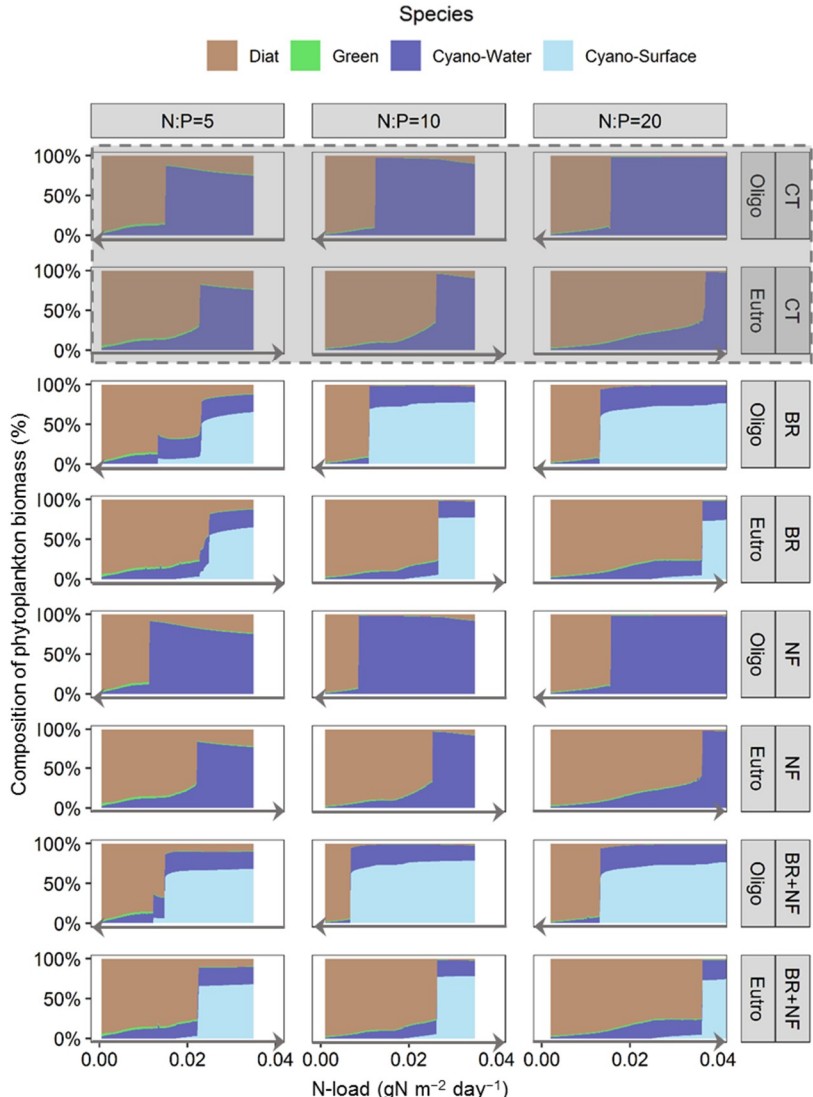

**Figure 3.** Composition of phytoplankton biomass in different scenarios during oligotrophication (Oligo) and eutrophication (Eutro) from bifurcation analyses. The grey box with dashed lines presents the results of the control (CT) scenario. Other scenarios shown in the figure are, from top to bottom, buoyancy regulation (BR), nitrogen fixation (NF), and a combination of buoyancy regulation and nitrogen fixation (BR + NF). Simulations started from a turbid initial state are marked by arrows towards the left; simulations started from a clear initial state are marked by arrows towards the right.

### 3.2. Effect of Nitrogen Fixation (NF Scenario)

As shown in Figure 1d–f, the addition of nitrogen fixation to the model increased the chl-*a* level when nitrogen was limited for phytoplankton, i.e., with an N:P ratio of 5 and 10. The chl-*a* level of the NF scenario overlapped with the CT scenario when the N:P ratio was 20 because sufficient nitrogen was available. The phytoplankton composition of the NF scenario was similar to the CT scenario with an exception around the position of nutrient loading thresholds. In both NF and CT scenarios, buoyancy regulation was disabled in the model simulations and the surface cyanobacteria were absent. Cyanobacteria in the water column were dominant when the lake was in a turbid state and diatoms were dominant within the phytoplankton community when the lake was in a clear state (Figure 3).

### 3.3. Synergistic Effect of Buoyancy Regulation and Nitrogen Fixation (BR + NF Scenario)

For the BR + NF scenario, the model showed a strong reduction of the nutrient loading thresholds during oligotrophication (Figure 1g–i). The triangles in Figure 2 showed that the combined effect of buoyancy regulation and nitrogen fixation on the nutrient loading thresholds was largest when the N:P ratio was 10, compared with all the other N:P ratios. As shown in Figure 2a, the difference between the nutrient loading thresholds of the CT scenario and the BR + NF scenario during oligotrophication was less than that of the NF scenario when the N:P ratio was 5, and identical to the BR scenario when the N:P ratio was 20. Like the BR scenario, the BR + NF scenario showed an intermediate state when the N:P ratio was 5 (Figure 1g). The intermediate state in the BR + NF scenario is shorter compared with the BR scenario. However, similar to the BR scenario when chl-*a* was at an intermediate level, macrophytes were absent (Figure S1) and diatoms were dominant (Figure 3). The chl-*a* level of the BR + NF scenario was higher than the chl-*a* level in the CT scenario when the N:P ratio was 5, but lower than the chl-*a* level in the CT scenario when the N:P ratio was either 10 or 20. Similarly to the BR scenario, in a turbid state, the chl-*a* level of the BR + NF scenario showed a more rapid increase than the chl-*a* level in the CT scenario as the nutrient loading increased. This led the chl-*a* level in the BR + NF scenario to surpass the chl-*a* level in the CT scenario when high nutrient loadings were applied in the simulations. Figure 3 shows that, under the BR + NF scenario, surface cyanobacteria were dominant when the lake was in a turbid state. This phytoplankton composition was comparable to the BR scenario.

## 4. Discussion

Our results obtained through bifurcation analyses with the expanded PCLake+ model indicate that the cyanobacterial traits of buoyancy regulation and nitrogen fixation reduce the value of nutrient loading thresholds, especially during oligotrophication. These effects on the nutrient loading thresholds differ between combinations of traits as captured in the four scenarios and with the limiting nutrients as expressed by the three N:P ratios. These results suggest that cyanobacterial traits can impede the effect of reducing nutrient loadings that are aiming to restore turbid lakes to their clear states. In contrast, small differences in nutrient loading thresholds between the control scenario and scenarios with traits suggest that cyanobacterial traits have a small effect on the shift in state during eutrophication.

### 4.1. Effect on Nutrient Loading Threshold

A reduction in nutrient loading threshold suggests an increase in the resistance of the turbid phytoplankton-dominated state to efforts of nutrient loading reduction. The underlying mechanisms of increasing resistance can be described with different feedback loops affected by cyanobacterial traits.

#### 4.1.1. Feedback Loops due to Buoyancy Regulation

The inclusion of buoyancy regulation results in a positive self-reinforcing feedback loop illustrated in Figure 4. In this positive feedback loop, the increase of cyanobacterial biomass results in lower light intensity but facilitates cyanobacterial growth even more. The effect of photoinhibition on surface cyanobacteria makes them less limited by light at high biomass and more limited by light at low biomass (Figure S2 in Supplementary material 4). As a result, cyanobacteria capable of buoyancy regulation are more resistant to nutrient reduction measures compared to cyanobacteria that do not have this trait. In comparison, cyanobacteria without buoyancy regulation (Figure 2, black line) usually experience a lower light intensity at which the increasing biomass will have a negative impact on itself, as shown by the negative feedback loop in Figure 4b.

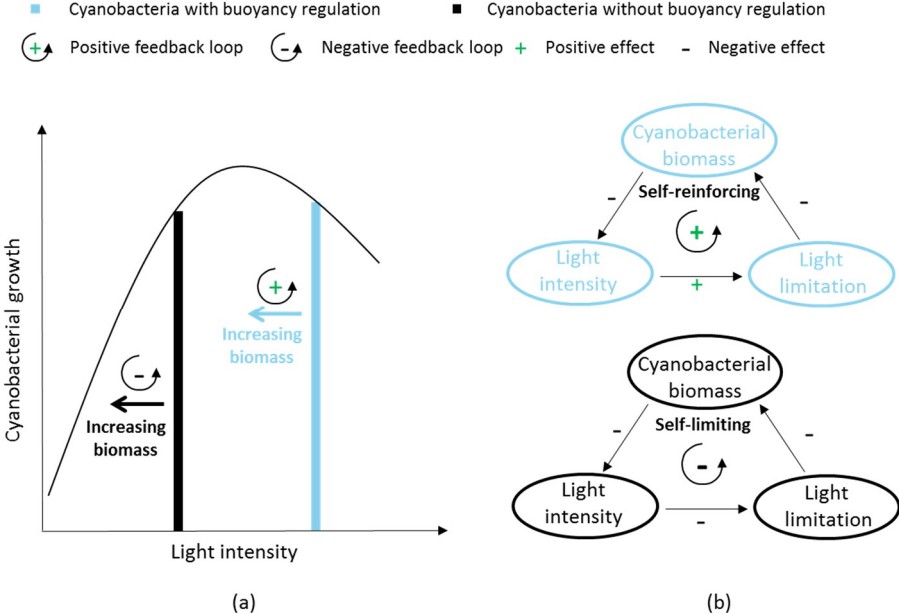

**Figure 4.** The relation between cyanobacterial growth and light intensity and the resulting feedback loop that self-reinforces or self-limits cyanobacterial biomass. (**a**) The cyanobacterial photosynthesis-irradiance (PI) curve showing the effect of light on cyanobacterial growth. The light blue bar is positioned at higher light intensity than the black bar because the cyanobacteria with buoyancy regulation can float to the surface and achieve a higher light intensity than the cyanobacteria staying in the water column. (**b**) The feedback loops of cyanobacterial biomass in response to light for the situation with (blue) and without buoyancy regulation (black).

## 4.1.2. Feedback Loops due to Nitrogen Fixation

Nitrogen fixation provides biologically available nitrogen to cyanobacteria, allowing them to achieve high levels of biomass [80]. In turn, higher cyanobacterial biomass increases the amount of nitrogen that can be fixed from dissolved $N_2$. As a result, nitrogen fixation leads to a positive feedback loop (Figure 5) from which cyanobacteria can benefit by becoming less nitrogen-limited compared to non-nitrogen fixers (Figure S3 in Supplementary material 4). Consequently, cyanobacteria that fix nitrogen are more resistant to nutrient-reduction measures compared to cyanobacteria that do not have this trait. This feedback loop provides most resistance in the nitrogen-limited system and is absent in the nitrogen sufficient system.

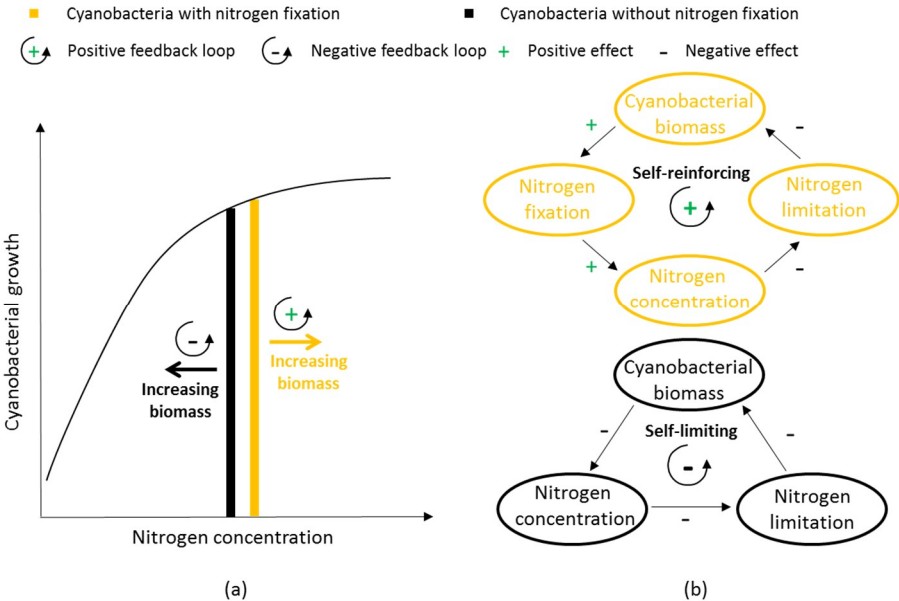

**Figure 5.** The relation between cyanobacterial growth and nitrogen concentration along with the resulting feedback loop that self-reinforces or self-limits cyanobacterial biomass. (**a**) The curve shows a positive relationship between nitrogen concentration and cyanobacterial growth. The yellow bar and the black bar are in a similar position since both types of cyanobacteria contain similar nitrogen concentrations in the water column. (**b**) The feedback loop of cyanobacteria with (yellow) and without nitrogen fixation (black).

### 4.1.3. Feedback Loops due to the Combination of the Two Traits

The impact of the combined cyanobacterial traits on nutrient loading threshold depends on the interplay of the two positive feedback loops of buoyancy regulation and nitrogen fixation (Figure 6). Buoyancy regulation enables more light availability for cyanobacteria so that the nitrogen fixation rate is maximized, thereby enhancing the nitrogen fixation feedback loop, as shown at the right in Figure 6. The increased nitrogen availability from nitrogen fixation supports growth of cyanobacteria and helps cyanobacteria approach the biomass needed to reduce photoinhibition, hence strengthening the feedback loop of the buoyancy regulation, as shown at the left in Figure 6. Consequently, cyanobacteria that have both traits are more resistant to nutrient-reduction measures compared to cyanobacteria that do not have any traits. Moreover, in real-world situations, cyanobacteria are known to exhibit a suite of other traits that make them more resistant to high light intensities at the surface (see e.g., [81,82]). Besides these two positive feedback loops, which are individually caused by buoyancy regulation and nitrogen fixation, the interplay between these two traits generate a third, negative, feedback loop (middle of Figure 6). This negative feedback loop causes biomass of cyanobacteria to decrease the light availability and lead to a reduction of nitrogen fixation rate, limiting the cyanobacterial biomass. This negative feedback loop is especially relevant in cases where a relatively small part of the cyanobacterial biomass is present at the surface layer; this small proportion of surface cyanobacteria still shades the cyanobacteria in the water column, leading to sub-optimal nitrogen fixation. We found that this negative feedback loop was present when the N:P ratio was 5. In this condition, cyanobacteria rely on extra N from nitrogen fixation to compensate for the nitrogen limitation but are unable to because of the lower light intensity (Figure S4). This interpretation is consistent with our results shown in Figure 2a that when N:P was 5, the single nitrogen fixation (NF scenario) has a larger effect on nutrient loading thresholds than the combination of two traits (BR + NF scenario).

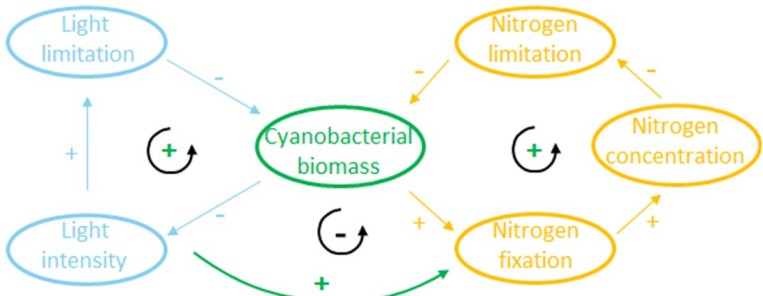

**Figure 6.** Feedback loop showing the combination of buoyancy regulation and nitrogen fixation. The relevant mechanisms of buoyancy regulation and nitrogen fixation are noted in blue and yellow, respectively. The interplay of the two functions is denoted in thick green arrows. The circular arrows denote positive or negative feedback loops, shown by a green plus sign and a black minus sign, respectively.

### 4.1.4. Effect on Lake Nutrient Loading Thresholds during Eutrophication

While these cyanobacterial traits reduced the nutrient loading thresholds during oligotrophication, they had little effect on determining the nutrient loading threshold during eutrophication. These nearly unaffected nutrient loading thresholds can be explained by the near absence of cyanobacteria in the clear macrophyte-dominated state. This can be explained from the positive feedback loops (Figures 4–6) that play a negligible role in the clear state. Also, the resistance of macrophytes to prevent a clear lake to become turbid due to eutrophication remains nearly unchanged. In contrast, the previous finding by Cottingham et al. [44] shows that cyanobacterial traits can reduce the resistance of a clear ecosystem to eutrophication. The model used by Cottingham et al. [44] is a simple conceptual model that lacks the effect of macrophytes and other food web components, whereas we simulated the nutrient loading thresholds with a process-based ecosystem model. The difference in model structures may explain the divergent conclusions of whether cyanobacterial traits can influence the resistance of a clear ecosystem to eutrophication.

### 4.2. Comparison to Natural Conditions

In contrast to our modelling study, most cyanobacterial blooms are a mixture of different types of cyanobacteria species that can fix nitrogen (e.g., *Nostoc* [31]), regulate buoyancy (e.g., *Microcystis* and *Oscillatoria* [27]), or do both (e.g., *Dolichospermum* and *Aphanizomenon* [5,83]). Their composition can be dynamic due to intraspecies competition under varying environmental conditions, such as light, temperature, and nutrients [84]. Here, we aimed to specify the effect of each cyanobacterial trait on determining the value of the nutrient loading thresholds. Our scenarios should, therefore, be seen as representative of functional groups in a case where one of the types of cyanobacterial species becomes dominant in the field. Simulations for intraspecies competition of cyanobacteria are outside the scope of this study.

Cyanobacteria with only buoyancy regulation usually outcompete cyanobacteria that have both nitrogen fixation and buoyancy traits (e.g., *Dolichospermum*) because they grow better under low light and use combined nitrogen more efficiently [85]. Moreover, nitrogen fixation is an energy-demanding process [80]. As a result of high energy costs, nitrogen-fixing cyanobacteria in a relatively N-rich environment have no competitive advantage, as shown in laboratory and field studies [86,87]. In a highly nitrogen-limited situation, however, nitrogen fixation can be advantageous to cyanobacteria. This is in line with our model results where we found that cyanobacteria with nitrogen fixation only affected the value of nutrient loading threshold in a nitrogen-limited or colimited environment (Figure 2). However, shallow lakes commonly appear to be phosphorus limited or colimited [88,89]. Our scenarios for phosphorus limited lakes show that cyanobacteria with buoyancy as traits have the highest effect on the nutrient loading threshold during oligotrophication. Under colimitation, our

results indicate that cyanobacteria with both traits have the largest impact on the nutrient loading threshold during oligotrophication (Figure 2).

Despite the capacity of cyanobacteria with buoyancy regulation to reduce the value of the nutrient loading thresholds during oligotrophication, our results also showed that their biomass during this transition is lower compared to cyanobacteria that do not have buoyancy regulation. Instead, these surface cyanobacteria obtained the highest biomass under hypereutrophic conditions where light is limiting (Figure 1). This can be explained by the photoinhibition caused by high light intensity on the surface that can only be reduced when surface blooms reach high biomass [90–92]. In reality, the photoinhibition caused by high light intensity may not be observed due to cyanobacterial adaptation [82] or high turbidity that lower the light intensity in the water column [93]. Besides, there is no spatial heterogeneity present in the surface layer in terms of cyanobacterial biomass (i.e., it is assumed to be a homogenous layer that is continually mixed). This results in all of the cyanobacteria in the surface layer negatively impacted by high light intensity due to photoinhibition. In the field, buoyant cyanobacteria are distributed heterogeneously along with depth, hence the cyanobacteria on the topmost surface can shelter other cyanobacteria from the high light intensity and eliminate photoinhibition [94]. Considering these findings, photoinhibition simulated in this study leads to a conservative prediction of cyanobacteria biomass compared with in situ conditions.

*4.3. Modelling as a Tool for Lake Managers*

The decrease in the value of nutrient loading thresholds due to the cyanobacterial traits, buoyancy regulation and nitrogen fixation, hinders attempts to improve water quality. Models such as PCLake+ can assist lake managers by calculating the nutrient loading threshold for lakes worldwide [17,40,95,96]. The tool we have developed here—an extension to the PCLake+ model—allows both scientists and managers to analyse a wide range of scenarios related to buoyancy regulation and nitrogen fixation, and allows for calculations of relevant nutrient loading limits at the local and catchment scale [97]. Besides, the result of our study can be applied in the remediation of eutrophic lakes because it shows at what N:P ratio the shallow lakes become either oligotrophic or eutrophic. Moreover, tools like PCLake+ can be used to study the effect of these traits on nutrient loading thresholds among a large variety of lakes by applying different lake characteristics (e.g., area and sediment type). Finally, the expanded PCLake+ model can be coupled with a hydrodynamic model to study cyanobacterial bloom dynamics in networks of water bodies or to predict the cyanobacteria scum in a spatially heterogeneous system (see e.g., Taihu [17]). Overall, this model enables scientists and managers to understand the underlying mechanism of lake resistance and to predict required nutrient reduction as the remedy to lakes suffering from cyanobacteria, thereby moving towards a sustainable future in the Anthropocene [98].

**5. Conclusions**

To mitigate cyanobacterial blooms, it is important to estimate the effects of cyanobacterial traits on the effort that is needed to reduce nutrient loadings. Here we used a modelling approach and found that nutrient loading thresholds of shallow lakes were lowered due to cyanobacterial buoyancy regulation and nitrogen fixation, especially during oligotrophication. Moreover, we explained the changes in nutrient loading thresholds by clarifying the positive feedback loops that involve buoyancy regulation and nitrogen fixation. These results help explain how these traits enhance the resistance of a lake's turbid, phytoplankton-dominated state to nutrient-reducing measures intended to achieve a clear macrophyte-dominated state. Our results point to the need to consider phytoplankton traits in lake management because they can hinder the effectiveness of measures to improve the water quality.

**Supplementary Materials:** The following are available online at http://www.mdpi.com/2073-4441/12/9/2467/s1, including the Conversion in Parameters of Nitrogen Fixation in Supplementary material 1, the R script for Bifurcation Analyses in Supplementary material 2, the expanded PCLake+ in Supplementary material 3, and Supplementary Figures in Supplementary material 4: Figure S1: Bifurcation plots of macrophyte dry weight, Figure S2: Light limitation of cyanobacteria from bifurcation analyses, Figure S3: Nutrient limitation of cyanobacteria from bifurcation analyses, Figure S4: Available light intensity of cyanobacteria from bifurcation analyses.

**Author Contributions:** Conceptualization, M.C., S.T., J.H.J., H.W.P. and A.B.G.J.; formal analysis, M.C.; methodology, M.C., S.T. and A.B.G.J.; project coordination, W.M.M.; supervision, S.T. and A.B.G.J.; writing—original draft preparation, M.C.; writing—review and editing, M.C., S.T., J.H.J., H.W.P., W.M.M. and A.B.G.J. All authors have read and agreed to the published version of the manuscript.

**Funding:** M.C. was supported by the Chinese Scholarship Council (CSC). S.T. was supported by Stichting Toegepast Onderzoek Waterbeheer (STOWA) (grant no. 443.269). A.B.G.J. is funded by Nederlandse Organisatie voor Wetenschappelijk Onderzoek (NWO) talent grant Veni with project number VI.Veni.194.002. H.W.P. was funded by the US National Science Foundation (1831096, 1840715). This is publication 7023 of the Netherlands Institute of Ecology (NIOO-KNAW).

**Acknowledgments:** We thank Dedmer van de Waal and Miquel Lurling for their contributions and sharing their knowledge on the physiology of cyanobacterial traits. We thank Maggie Armstrong for the contributions on checking and revising the entire manuscript for English writing.

**Conflicts of Interest:** The authors declare no conflict of interest. The funders had no role in the design of the study; in the collection, analyses, or interpretation of data; in the writing of the manuscript, or in the decision to publish the results.

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
