# Peer review of "Exploring How Cyanobacterial Traits Affect Nutrient Loading Thresholds in Shallow Lakes: A Modelling Approach"

_water, doi:10.3390/w12092467_

Round 1

Reviewer 1 Report

This study focuses on cyanobacteria in shallow lakes and critical nutrient loadings where lake clarity shifts from clear to turbid or vice versa.  The topic is highly relevant, especially today as climate change is impacting temperature regimes and theprevalence of hazardous algal blooms (HABS). The approach taken is imaginative and the use of graphics with appropriate explanation is well done.  There are however major deficiencies in this study than need to be addressed.

1.  The most serious problem is poor english grammar and lack of subject object correspondence. For example the phrases: "lakes that accommodate cyanobacteria with these traits show increased resistance to restoration and thus need more restoration effort" "cyanobacteria that fix nitrogen put more resistance to a decreasing nutrient loading" "these cyanobacterial traits reduce the critical nutrient loading from turbid to clear".  In all cases the subject of the sentence - the lake, cyanobacteria or cyanobacterial traits do not match the verb. Cyanobacteria to do resist restoration or nutrient loading. Cyanobacterial traits may affect nutrient loading but nutrient loading is not measured in units of turbid or clear.  these are seconady impacts.  This problem has been highlighted in the attached annotated text.  Every instance needs to be addressed before acceptance for publication.

2.  The manuscript is repetitive in many sections and can be considerably shortened if this repetition is eliminated.  Organization is good but this tendency to repeat statements detracts from the quality of the presentation.

3.  I have flagged several of the figure captions as been excessively long.  In general I am in favor of descriptive captions that explain clearly want is being shown in the figure.  It avoids asking the reader to flip back and forth between the manuscript text and figure.  However a caption that consumes more than 1/2 page is unusual and may be flagged by the journal production department if this paper is finally accepted.  Try to consolidate these captions to 4-5 sentences max.

4.  This is a very simple modeling study and the study would benefit from the inclusion of experimental field data. The conclusions reached in this study are not suprising and can be intuited by most limnologists or environmental scientists.  The study is not that profound and I am not sure makes a significant contribution to the science. However the authors have done a decent job in explaining the limitations of an existing model and how it has been enhanced through the inclusion of the bifurcation analyses.

5.  The authors use a lot of jargon in this paper as well as colloquial expressions that have no value in scientific writing.  The bifurcation analysis is an important aspect of this study and should be explained in detail. Other analyses and sub-models such as Michaelis-Menten kinetic expressions should also be explained. 

Reviewer 2 Report

web submission prob encountered

Round 2

Reviewer 1 Report

The authors have addressed my major concerns.  No further comments.